# AV-Deepfake1M: A Large-Scale LLM-Driven Audio-Visual Deepfake Dataset

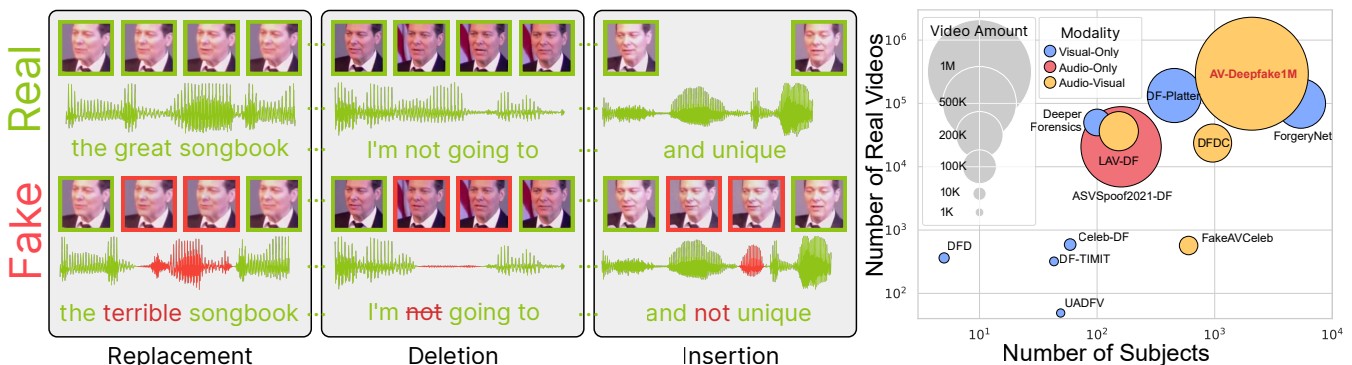

**Figure 1: AV-Deepfake1M is a large-scale content-driven deepfake dataset generated by utilising a large language model.** The dataset contains more than 2K subjects and 1M deepfake videos generated by employing different audio-visual content manipulation strategies. The left figure illustrates examples of word-level *replacement*, *deletion*, and *insertion* strategies to manipulate audio-visual content. The right figure illustrates a comparison between the proposed dataset and other publicly available datasets in terms of the number of subjects, and amount of real and fake videos.

## ABSTRACT

The detection and localization of highly realistic deepfake audio-visual content are challenging even for the most advanced state-of-the-art methods. While most of the research efforts in this domain are focused on detecting high-quality deepfake images and videos, only a few works address the problem of the localization of small segments of audio-visual manipulations embedded in real videos. In this research, we emulate the process of such content generation and propose the AV-Deepfake1M dataset. The dataset contains content-driven (i) video manipulations, (ii) audio manipulations, and (iii) audio-visual manipulations for more than 2K subjects resulting in a total of more than 1M videos. The paper provides a thorough description of the proposed data generation pipeline accompanied by a rigorous analysis of the quality of the generated data. The comprehensive benchmark of the proposed dataset utilizing state-of-the-art deepfake detection and localization methods indicates a significant drop in performance compared to previous datasets. The proposed dataset will play a vital role in building the next-generation deepfake localization methods. The dataset and associated code will be made public.

**Unpublished working draft. Not for distribution.**

## CCS CONCEPTS

• **Computing methodologies** → Computer vision; Machine learning approaches.

## KEYWORDS

Datasets, Deepfake, Localization, Detection

## 1 INTRODUCTION

We are witnessing rapid progress in the domain of content generation technology, i.e., models trained on massive amounts of data that can produce highly realistic text [3, 51, 52], video [18, 49, 59] and audio [27, 28, 45]. Consequently, discriminating between real and fake content is becoming increasingly more challenging even for humans [38, 67]. This opens the door for misuse of content generation technology for example to spread misinformation and commit fraud, rendering the development of reliable detection methods vital.

The development of such methods is highly dependent on the available deepfake benchmark datasets, which led to the steady increase in the number of publicly available datasets that provide examples of visual-only [26, 33, 36], audio-only [37, 62], and audio-visual [29] content modification strategies (e.g., face-swapping, face-reenactment, etc.). However, the majority of these datasets and methods assume that the entirety of the content (i.e., audio, visual, audio-visual) is either real or fake. This leaves the door open for criminals to exploit the embedding of small segments of manipulations in the otherwise real content. As argued in [6], this type of targeted manipulation can lead to drastic changes in the underlying meaning as illustrated in Figure 1. Given that most deepfake benchmark datasets do not include this new type of manipulation strategy, state-of-the-art

**Table 1: Details for publicly available deepfake datasets in a chronologically ascending order.** Cla: Binary classification, SL: Spatial localization, TL: Temporal localization, FS: Face swapping, RE: Face reenactment, TTS: Text-to-speech, VC: Voice conversion.

| Dataset | Year | Tasks | Manipulated Modality | Manipulation Method | #Subjects | #Real | #Fake | #Total |
|---------|------|-------|---------------------|---------------------|-----------|-------|-------|--------|
| DF-TIMIT [32] | 2018 | Cla | V | FS | 43 | 320 | 640 | 960 |
| UADFV [61] | 2019 | Cla | V | FS | 49 | 49 | 49 | 98 |
| FaceForensics++ [44] | 2019 | Cla | V | FS/RE | - | 1,000 | 4,000 | 5,000 |
| Google DFD [39] | 2019 | Cla | V | FS | 5 | 363 | 3,068 | 3,431 |
| DFDC [16] | 2020 | Cla | AV | FS | 960 | 23,654 | 104,500 | 128,154 |
| DeeperForensics [26] | 2020 | Cla | V | FS | 100 | 50,000 | 10,000 | 60,000 |
| Celeb-DF [36] | 2020 | Cla | V | FS | 59 | 590 | 5,639 | 6,229 |
| WildDeepfake [68] | 2020 | Cla | - | - | - | 3,805 | 3,509 | 7,314 |
| FFIW$_{10K}$ [67] | 2021 | Cla/SL | V | FS | - | 10,000 | 10,000 | 20,000 |
| KoDF [33] | 2021 | Cla | V | FS/RE | 403 | 62,166 | 175,776 | 237,942 |
| FakeAVCeleb [29] | 2021 | Cla | AV | RE | 600+ | 570 | 25,000+ | 25,500+ |
| ForgeryNet [21] | 2021 | SL/TL/Cla | V | Random FS/RE | 5,400+ | 99,630 | 121,617 | 221,247 |
| ASVSpoof2021DF [37] | 2021 | Cla | A | TTS/VC | 160 | 20,637 | 572,616 | 593,253 |
| LAV-DF [6] | 2022 | TL/Cla | AV | Content-driven RE/TTS | 153 | 36,431 | 99,873 | 136,304 |
| DF-Platter [38] | 2023 | Cla | V | FS | 454 | 133,260 | 132,496 | 265,756 |
| AV-Deepfake1M | 2023 | TL/Cla | AV | Content-driven RE/TTS | 2,068 | 286,721 | 860,039 | 1,146,760 |

detection methods might fail to perform reliably on this new type of deepfake content.

This work addresses this gap by releasing a new large-scale audio-visual dataset called AV-Deepfake1M specifically designed for the task of temporal deepfake localization. To improve the realism and quality of generated content, the proposed data generation pipeline incorporates the ChatGPT[1] large language model. The pipeline further utilizes the latest open-source state-of-the-art methods for high-quality audio [8, 31] and video [54] generation. The scale and novel modification strategies position the proposed dataset as the most comprehensive audio-visual benchmark as illustrated in Figure 1, making it an important asset for building the next generation of deepfake localization methods. The main contributions of this work are,

- We propose AV-Deepfake1M, a large-scale content-driven audio-visual dataset for the task of temporal deepfake localization.
- We propose an elaborate data generation pipeline employing novel manipulation strategies and incorporating the state-of-the-art in text, video and audio generation.
- We perform comprehensive analysis and benchmark of the proposed dataset utilizing state-of-the-art deepfake detection and localization methods.

## 2 RELATED WORK

The performance of any deepfake detection method is highly dependent on the quantitative and qualitative aspects of the datasets used for development. Over the past few years, many datasets (e.g., [21, 32, 38]) have been proposed to support the research on deepfake detection. A comprehensive list of the relevant publicly available datasets is given in Table 1. Most of the available datasets provide examples of face manipulations through either face swapping [16, 32, 67] or face reenactment [29, 33] techniques. In terms of the number of samples, earlier datasets are smaller due to the

[1]https://chat.openai.com/

limited availability of face manipulation techniques. With the rapid advancements in content generation technology, several large-scale datasets such as DFDC [16], DeeperForensics [26], KoDF [33], and DF-Platter [38] have been proposed. However, the task associated with these datasets is mainly restricted to coarse-level deepfake detection. Until this point manipulations are mainly applied only to the visual modality, and later, audio manipulations [37] and audio-visual manipulations [29] have been proposed to increase the complexity of the task.

In 2022, LAV-DF [6] was introduced to become the first content-driven deepfake dataset for temporal localization. However, the quality and scale of LAV-DF are limited, and the state-of-the-art methods designed for temporal localization [4, 65] are already achieving very strong performance. AV-Deepfake1M addresses these gaps by improving the quality, diversity, and scale of the previous datasets designed for temporal deepfake localization. Given that LAV-DF is the only available public dataset that has been designed for the same task as the dataset proposed in this paper, next we do a direct comparison of the two datasets. In addition to the fact that AV-Deepfake1M is significantly larger than LAV-DF, in terms of the number of subjects, and amount of real and fake videos, the following differences further highlight our contributions.

- LAV-DF uses a rule-based system to find antonyms that maximize the change in sentiment in the transcript manipulation step. We argue that naively choosing the antonyms causes context inconsistencies and low diversity of the fake content. AV-Deepfake1M addresses this issue with the use of a large language model, which results in diverse and context-consistent fake content.
- The output quality of the visual generator Wav2Lip [42] and audio generator SV2TTS [25] used for generating LAV-DF is not sufficient for state-of-the-art detection methods. AV-Deepfake1M utilizes the latest open-source state-of-the-art methods for high-quality audio and video generation.

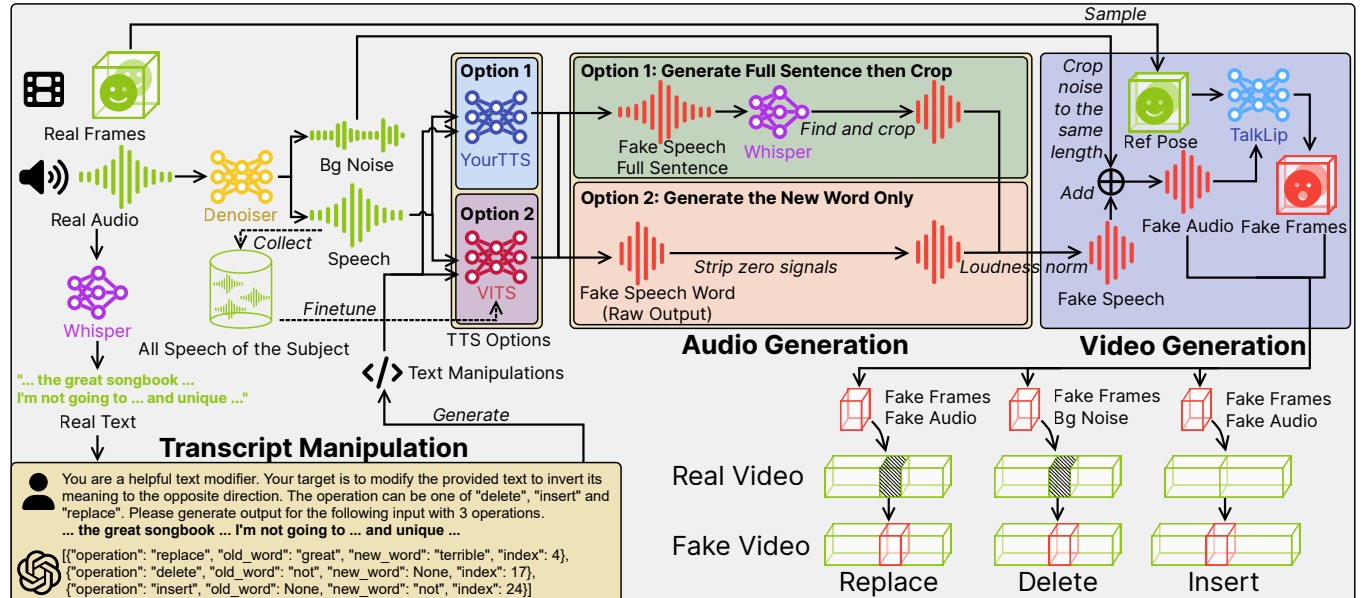

**Figure 2: Data manipulation and generation pipeline.** Overview of the proposed three-stage pipeline. Given a real video, the pre-processing consists of audio extraction via FFmpeg followed by Whisper-based transcript generation. In the first stage, transcript manipulation, the original transcript is modified through word-level insertions, deletions, and replacements. In the second stage, audio generation, based on the relevant transcript manipulation, the audio is generated in both speaker-dependent and independent fashion. In the final stage, video generation, based on the generated audio, the subject-dependant video is generated with smooth transitions in terms of lip-synchronization, pose, and other relevant attributes.

- LAV-DF includes only *replacement* as a manipulation strategy. AV-Deepfake1M includes two additional challenging manipulation strategies, *deletion* and *insertion*.

## 3 AV-DEEPFAKE1M DATASET

AV-Deepfake1M is a large-scale audio-visual deepfake dataset, including 1,886 hours of audio-visual data from 2,068 unique subjects captured in diverse background environments. This positions the proposed dataset as the most comprehensive audio-visual benchmark as illustrated in Figure 1 and Table 1. The generated videos in AV-Deepfake1M preserve the background and identity present in the real videos, while the content is carefully manipulated with content-driven audio-visual data. Following previous deepfake dataset generation research [6, 29], the dataset includes three different combinations of modified modalities in the generated fake videos. Please note that here we also introduce the concept of content-driven modifications for unimodal as well as multimodal aspects. We further elaborate on this in the supplementary material.

- **Fake Audio** and **Fake Visual.** Both the real audio and visual frames are manipulated.
- **Fake Audio** and **Real Visual.** Only the real audio corresponding to *replacements* and *deletions* is manipulated. To further increase data quality, the fake audio, and the corresponding length-normalized real visual segments are synchronized. As for the *insertions*, new visual segments are generated based on the length of the fake audio and are lip-synced to the background noise (i.e., closed mouth).

- **Real Audio** and **Fake Visual.** Only the real visual frames corresponding to *replacements* and *deletions* are manipulated. To further increase data quality, the length of the fake visual segments is normalized to match the length of the real audio. As for the *insertions*, background noise is inserted for the corresponding fake visual segments.

### 3.1 Data Generation Pipeline

The three-stage pipeline for generating content-driven deepfakes is illustrated in Figure 2. A subset of real videos from the Voxceleb2 [14] dataset is pre-processed to extract the audio using FFmpeg [50], followed by Whisper-based [43] real transcript generation.

#### 3.1.1 Transcript Manipulation.

**Manipulation Strategy.** The first stage for generating content-driven deepfakes is transcript manipulation. We utilize ChatGPT for altering the real transcripts. Through LangChain [9] the output of ChatGPT is a structured JSON with four fields: 1) `operation:` This set contains *replace*, *delete*, and *insert*, which has been applied on the input; 2) `old_word:` The word in the input to *replace* or *delete*; 3) `new_word:` The word in the input to *insert* or *replace*; 4) `index:` The location of the operation in the input. The number of transcript modifications depends on the video length and is determined by the following equation $M = \mathbf{ceil}(t/10)$ where $M$ is the number of modifications and $t$ (sec) is the length of the video. We followed [3] and built a few-shot prompt template for ChatGPT.

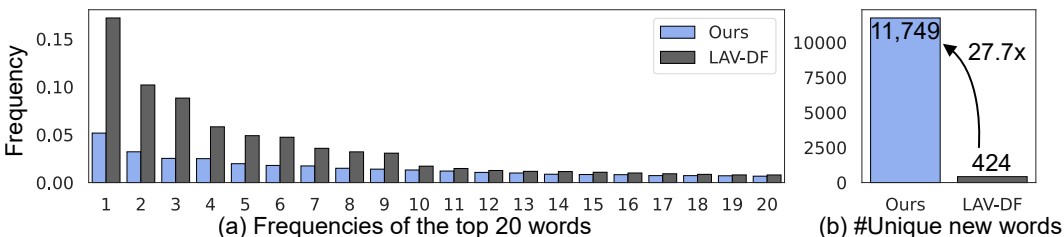

Figure 3: Comparison of transcript modifications in AV-Deepfake1M and LAV-DF.



> **Prompt 3.1: Transcripts manipulation**
>
> **System:** You are a helpful text modifier. Your target is to modify the provided text to invert its meaning to the opposite direction. Here is the transcript of the audio. Please use the provided operations to modify the transcript to change its meaning. The operation can be one of "delete", "insert" and "replace".
> **Human:** {EXAMPLE INPUT 1}
> **AI:** {EXAMPLE OUTPUT 1}
> **Human:** {EXAMPLE INPUT 2}
> **AI:** {EXAMPLE OUTPUT 2}
> ......
> **Human:** Please generate output for the following input with {NUM} operations. {INPUT}

**Analysis.** Figure 3 (a) illustrates a comparison of the frequencies of the top 20 words in AV-Deepfake1M and LAV-DF [6]. The results show that few words in LAV-DF have dominant frequencies (> 10%), whereas this issue is drastically reduced in AV-Deepfake1M. Owing to the contribution of ChatGPT, we also observed a significant increase in unique new words (27.7 times more) in the modified transcripts compared to LAV-DF, illustrated in Figure 3 (b). This statistical comparison shows that the proposed LLM-based transcript manipulation strategy generates more diverse content compared to the rule-based strategy employed in LAV-DF. We further elaborate on the advantages of using an LLM in this step in the supplementary material.

### 3.1.2 Audio Generation.

**Manipulation Strategy.** The next stage is to generate high-quality audio with the same style as the speaker. The audio is first separated into background noise and speech using Denoiser [17]. Zero-shot voice cloning methods such as SV2TTS [25] utilized by previous datasets [6, 29] have low signal-to-noise ratio resulting in low-quality audio manipulations that are easily localized by BA-TFD [4] and UMMAFormer [65]. To increase the quality of the generated audio, we employ the identity-dependent text-to-speech method VITS [31] for a subset of the subjects. Further diversity in the audio generation was introduced by utilizing the identity-independent text-to-speech method YourTTS [8] for the rest of the subjects.

Audio generation is slightly different for each of the manipulation strategies (i.e., *replace*, *insert* and *delete*). In the case of *replace* and *insert*, we need to generate new audio corresponding to `new_word`(s). Generally, there are two ways to generate the `new_word`(s): 1) Generate audio for the final fake transcript and crop it to get the audio for the required `new_word`(s) and 2) Generate audio only for the `new_word`(s). To bring further diversity and challenge, we use both strategies to generate audio for the `new_word`(s). In the case of *delete*, only the background noise is retained. After the audio manipulation, we normalized the loudness

**Table 2: Audio quality of AV-Deepfake1M.** Quality of the generated audio in terms of SECS, SNR and FAD.

| Dataset | SECS(↑) | SNR(↑) | FAD(↓) |
|---|---|---|---|
| FakeAVCeleb [29] | 0.543 | 2.16 | 6.598 |
| LAV-DF [6] | 0.984 | 7.83 | 0.306 |
| AV-Deepfake1M (Train) | 0.991 | 9.40 | 0.091 |
| AV-Deepfake1M (Validation) | 0.991 | 9.16 | 0.091 |
| AV-Deepfake1M (Test) | 0.991 | 9.42 | 0.083 |
| AV-Deepfake1M (Overall) | **0.991** | **9.39** | **0.088** |

of the fake audio segments to the original audio to add more realism. Finally, to keep the consistency with the environmental noise, we add the background noise previously separated to the final audio output.

**Analysis.** We evaluated the quality of the audio generation following previous works [7, 11] (note that for all datasets, we only evaluated the samples where the audio modality is modified). The results are shown in Table 2. The first evaluation metric is speaker encoder cosine similarity (SECS) [53]. It measures the similarity of the speakers given a pair of audio in the range $[-1, 1]$. We also calculated the signal-to-noise ratio (SNR) for all fake audio and Fréchet audio distance (FAD) [30]. The results indicate that AV-Deepfake1M contains higher quality audio compared to other datasets.

### 3.1.3 Video Generation.

**Manipulation Strategy.** The final stage of the generation pipeline is visual content generation. After the audio is generated, the lip-synced visual frames are generated based on the subjects' original pose and the fake audio. We investigated several face reenactment strategies including EAMM [24], AVFR-GAN [2], DiffTalk [46], AD-NeRF [19] and ATVGnet [10] and concluded that these methods are not well suited for zero-shot lip-synced generation of unseen speakers. Thus, we use TalkLip [54] for visual content generation which is primarily designed for zero-shot lip-sync scenarios. LipTalk is 1) Identity-independent, 2) Lip-syncing only without generating new poses, 3) Fast, 4) State-of-the-art, and 5) Open-source. This way we avoid the weaknesses of the aforementioned face reenactment strategies. The pre-trained TalkLip model is used to generate fake visual frames that are lip-synchronized with the input audio and can be used for *insertion*, *replacement*, and *deletion*.

**Analysis.** To evaluate the visual quality of the proposed dataset, we used peak signal-to-noise ratio (PSNR), structural similarity

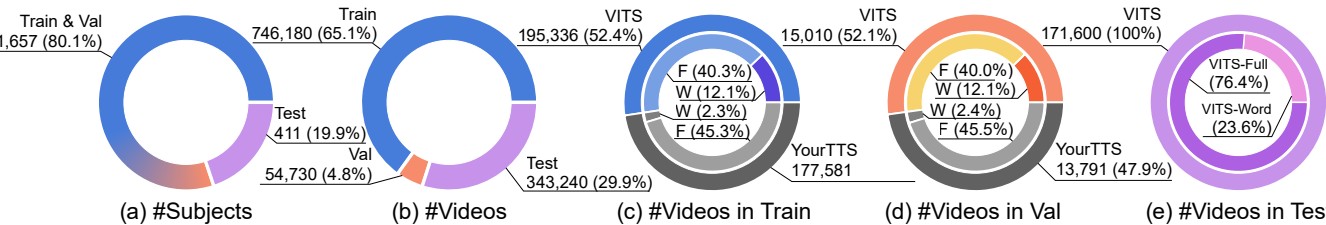

**Figure 4: Data partitioning in AV-Deepfake1M.** (a) The number of subjects in the *train*, *validation*, and *test* sets. (b) The number of videos in the *train*, *validation*, and *test* sets. (c) The number of videos with different audio generation methods in the *train* set. (d) The number of videos with different audio generation methods in the *validation* set. (e) The number of videos with different audio generation methods in the *test* set. In (c, d, e), F denotes audio generation for the *full* transcript and cropping of the `new_word`(s) while W denotes audio generation only for the `new_word`(s).

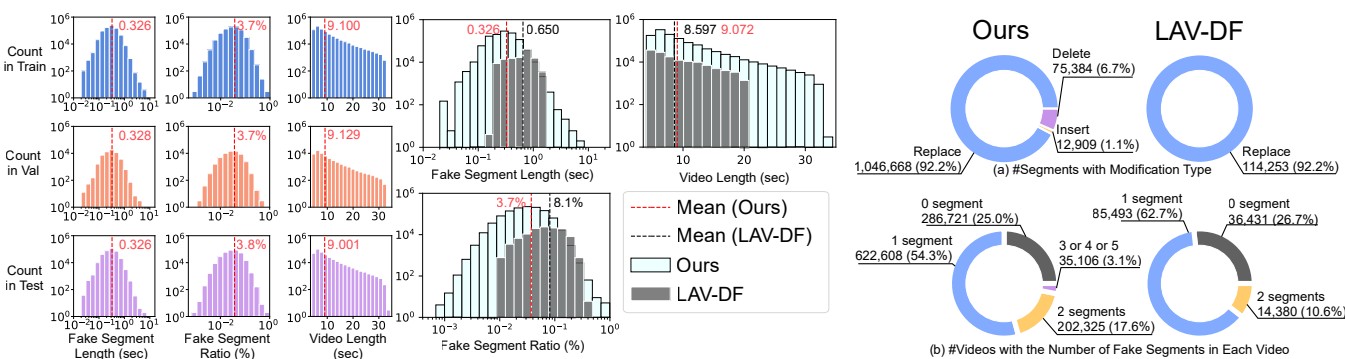

**Figure 5: Comparison of AV-Deepfake1M and LAV-DF.** The left three-row three-column histograms illustrate the fake segment absolute lengths (sec), the fake segment lengths proportion in videos (%) and the video lengths (sec) in the *train*, *validation*, and *test* sets. In the middle, the histograms illustrate the overall statistics for fake segment lengths, proportions and video lengths, compared with LAV-DF. For the fake segment lengths and proportions, the X-axis is in log scale and for video lengths, the X-axis is in linear scale. For all histograms, the Y-axis is in linear scale. The vertical dotted lines and numbers in histograms represent the mean value. On the right side, (a) The number of segments with different modifications and (b) The number of videos with different numbers of segments per video.

**Table 3: Visual quality of AV-Deepfake1M.** Quality of the generated video in terms of PSNR, SSIM and FID.

| Dataset | PSNR(↑) | SSIM(↑) | FID(↓) |
|---|---|---|---|
| FF++ [44] | 24.40 | 0.812 | 1.06 |
| DFDC [16] | - | - | 5.69 |
| FakeAVCeleb [29] | 29.82 | 0.919 | 2.29 |
| LAV-DF [6] | 33.06 | 0.898 | 1.92 |
| AV-Deepfake1M (Train) | 39.50 | 0.977 | 0.50 |
| AV-Deepfake1M (Validation) | 39.54 | 0.977 | 0.49 |
| AV-Deepfake1M (Test) | 39.48 | 0.977 | 0.56 |
| AV-Deepfake1M (Overall) | **39.49** | **0.977** | **0.49** |

index (SSIM) [58] and Fréchet inception distance (FID) [23] metrics as shown in Table 3. Note that for a fair comparison, we preprocessed the videos to a common format. The videos of FF++ [44] and DFDC [16] are 'in-the-wild', whereas FakeAVCeleb [29], LAV-DF [6] and AV-Deepfake1M are facial videos. Thus, we cropped the facial region for FF++ and DFDC for visual quality assessment. Since FakeAVCeleb, LAV-DF and AV-Deepfake1M are multimodal, for a fair comparison, we only used the samples with the visual

modality modified to compute the metrics. The results indicate that AV-Deepfake1M is of higher visual quality compared to existing datasets.

## 3.2 Dataset Statistics

We split the dataset into *train*, *validation*, and *test* sets. We first randomly select 1,657 subjects for the *train* set and 411 subjects for the *test* set without any overlap. The *validation* set is selected randomly from the *train* subset. The *test* set contains only samples with VITS-based identity-dependent audio. The variation in the number of subjects and videos in different sets is presented in Table 4 and Figure 4.

Figure 5 illustrates the direct comparison of AV-Deepfake1M and LAV-DF [6]. The results indicate that AV-Deepfake1M is more diverse in terms of modifications, subjects, fake segment and video lengths, and a lower average proportion of fake segments, making the dataset a vital asset for building better deepfake localization methods.

**Table 4: Number of subjects and videos in AV-Deepfake1M.**

| Subset | #Subjects | #Real Videos | #Fake Videos | #Videos |
|--------|-----------|--------------|--------------|---------|
| Train | 1,657 | 186,666 | 559,514 | 746,180 |
| Validation | | 14,235 | 43,105 | 54,730 |
| Test | 411 | 85,820 | 257,420 | 343,240 |
| Overall | 2,068 | 286,721 | 860,039 | 1,146,760 |

**Table 5: User study results for AV-Deepfake1M and LAV-DF.**

| User Study | Acc. | AP@0.1 | AP@0.5 | AR@1 |
|-----------|------|--------|--------|------|
| LAV-DF | 84.03 | 36.80 | 14.17 | 10.04 |
| AV-Deepfake1M | 68.64 | 15.32 | 01.92 | 02.54 |

### 3.3 Human Quality Assessment

To investigate if humans can detect the deepfakes in AV-Deepfake1M, we also conducted a user study with 25 participants with prior experience in video manipulation in the computer vision domain (note that the authors did not participate in the study). 200 random samples that contain 0 or 1 modification were selected for the study, where 100 from LAV-DF and 100 from AV-Deepfake1M. Each participant was asked to classify 20 videos (5 real and 5 fake from LAV-DF dataset, 5 real and 5 fake from AV-Deepfake1M) as real or fake and propose the potential fake segment start and end point. The user study results presented in Table 5 indicate that the deepfake content in AV-Deepfake1M is very challenging to detect for humans, and AV-Deepfake1M is more difficult than LAV-DF.

### 3.4 Computational Cost

We spent around ~600 GPU hours for speech recognition with Whisper [43], ~2100 GPU hours for training VITS [31] (each of the 721 VITS models requires ~3hrs), and ~300 GPU hours for data generation. Overall, we needed ~3000 GPU hours to generate AV-Deepfake1M with NVIDIA RTX6000 GPUs.

## 4 BENCHMARKS AND METRICS

This section outlines the benchmark protocol for AV-Deepfake1M along with the used evaluation metrics. The goal is to detect and localize content-driven audio, visual, and audio-visual manipulations.

### 4.1 Data Partitioning

The dataset is organized in *train*, *validation*, and *test* sets, as described in Section 3.2. The original *test* set (all modifications) is referred to as *fullset* in the rest of the text. For a fair comparison with visual-only and audio-only methods, we also prepared *subset V* (by excluding the videos with audio-only modifications from *fullset*) and *subset A* (by excluding the videos with visual-only modifications from *fullset*).

### 4.2 Implementation Details

For benchmarking temporal deepfake localization, we consider the following state-of-the-art methods: Pyannote [41] is a pre-trained speaker diarization method. TriDet [47] and ActionFormer [63] are the state-of-the-art in the temporal action localization domain. Since these two methods require pre-trained features, we extracted the

state-of-the-art features VideoMAEv2 [56] and InternVideo [57] for benchmarking. BA-TFD [6], BA-TFD+ [4], and UMMAFormer [65] are the state-of-the-art methods specifically designed for audio-visual temporal deepfake localization. We followed the original settings for BA-TFD and BA-TFD+. For UMMAFormer [65], we implemented it using the InternVideo [57] visual features and BYOL-A [40] audio features. For image-based classification methods, we consider Meso4 [1], MesoInception4 [1], Xception [12], Face X-Ray [34], LipForensics [20], EfficientViT [15], and SBI [48]. We followed the procedure used in previous works [4, 66] to aggregate the frame-level predictions to segments for localization.

For benchmarking deepfake detection, we trained the image-based models Meso4 [1], MesoInception4 [1], Xception [12] and EfficientViT [15] with video frames along with the corresponding labels. For the segment-based methods MDS [13] and MARLIN [5], we used a sliding window to sample segments from the video for training and inference. During the inference stage, the frame- and segment-level predictions are aggregated to video-level by *max* voting. The aggregation strategy is discussed in Section 5. We also evaluated the zero-shot performance of several methods, including the LLM-based Video-LLaMA [64], audio pre-trained CLAP [60], M2TR [55] and LipForensics [20] pre-trained on FF++ [44], Face X-Ray [34] and SBI [48] pretrained on blending images. For Video-LLaMA, we also evaluated 5 model ensembles (the majority vote of 5 model inferences). To investigate the impact of the level of label access, we designed 3 different label access levels for training: *frame-level* labels, *segment-level* labels only, and *video-level* labels only.

### 4.3 Evaluation Metrics

**Temporal Deepfake Localization.** We use average precision (AP) and average recall (AR) as prior works [6, 21].
**Deepfake Detection.** We use the standard evaluation protocol [16, 44] and report video-level accuracy (Acc.) and area under the curve (AUC).

## 5 RESULTS AND ANALYSIS

This section reports the performance of the state-of-the-art deepfake detection and localization methods described in Section 4.2 on AV-Deepfake1M. The reported performance is based on different subsets, described in Section 4.1, and different levels of label access during training, described in Section 4.2.

### 5.1 Audio-Visual Temporal Deepfake Localization

The results of this benchmark are depicted in Table 6. All state-of-the-art methods achieve significantly lower performance compared to the performance reported on previous datasets [6, 21]. This significant drop indicates that existing temporal deepfake localization methods are falling behind with the rapid advancements in content generation. In other words, we can claim that the highly realistic fake content in AV-Deepfake1M will open an avenue for further research on temporal deepfake localization methods.

### 5.2 Audio-Visual Deepfake Detection

Similarly to temporal deepfake localization, the results of the classical deepfake detection benchmark are summarized in Table 7.

**Table 6: Temporal deepfake localization benchmark.** Performance comparison of state-of-the-art methods on the proposed AV-Deepfake1M dataset. The results are significantly low, indicating that AV-Deepfake1M is an important benchmark for this task.

| Set | Method | Mod. | AP@0.5 | AP@0.75 | AP@0.9 | AP@0.95 | AR@50 | AR@30 | AR@20 | AR@10 | AR@5 |
|---|---|---|---|---|---|---|---|---|---|---|---|
| Fullset | PyAnnote (Zero-Shot) [41] | A | 00.03 | 00.00 | 00.00 | 00.00 | 00.67 | 00.67 | 00.67 | 00.67 | 00.67 |
| | Meso4 [1] | V | 09.86 | 06.05 | 02.22 | 00.59 | 38.92 | 38.91 | 38.81 | 36.47 | 26.91 |
| | MesoInception4 [1] | V | 08.50 | 05.16 | 01.89 | 00.50 | 39.27 | 39.22 | 39.00 | 35.78 | 24.59 |
| | EfficientViT [15] | V | 14.71 | 02.42 | 00.13 | 00.01 | 27.04 | 26.99 | 26.43 | 23.90 | 20.31 |
| | TriDet + VideoMAEv2 [47, 56] | V | 21.67 | 05.83 | 00.54 | 00.06 | 20.27 | 20.23 | 20.12 | 19.50 | 18.18 |
| | TriDet + InternVideo [47, 57] | V | 29.66 | 09.02 | 00.79 | 00.09 | 24.08 | 24.06 | 23.96 | 23.50 | 22.55 |
| | ActionFormer + VideoMAEv2 [56, 63] | V | 20.24 | 05.73 | 00.57 | 00.07 | 19.97 | 19.93 | 19.81 | 19.11 | 17.80 |
| | ActionFormer + InternVideo [57, 63] | V | 36.08 | 12.01 | 01.23 | 00.16 | 27.11 | 27.08 | 27.00 | 26.60 | 25.80 |
| | BA-TFD [6] | AV | 37.37 | 06.34 | 00.19 | 00.02 | 45.55 | 40.37 | 35.95 | 30.66 | 26.82 |
| | BA-TFD+ [4] | AV | 44.42 | 13.64 | 00.48 | 00.03 | **48.86** | **44.51** | 40.37 | 34.67 | 29.88 |
| | UMMAFormer [65] | AV | **51.64** | **28.07** | **07.65** | **01.58** | 44.07 | 43.93 | **43.45** | **42.09** | **40.27** |
| Subset V | PyAnnote (Zero-Shot) [41] | A | 00.02 | 00.00 | 00.00 | 00.00 | 00.52 | 00.52 | 00.52 | 00.52 | 00.52 |
| | Meso4 [1] | V | 15.31 | 09.54 | 03.52 | 00.93 | 58.04 | 58.03 | 57.87 | 54.37 | 40.06 |
| | MesoInception4 [1] | V | 13.38 | 08.25 | 03.05 | 00.81 | 58.54 | 58.48 | 58.15 | 53.34 | 36.59 |
| | EfficientViT [15] | V | 23.21 | 03.92 | 00.21 | 00.02 | 37.52 | 37.46 | 36.88 | 34.19 | 29.64 |
| | TriDet + VideoMAEv2 [47, 56] | V | 26.45 | 07.35 | 00.74 | 00.08 | 22.49 | 22.47 | 22.42 | 22.04 | 21.09 |
| | TriDet + InternVideo [47, 57] | V | 37.90 | 12.15 | 01.12 | 00.13 | 28.08 | 28.07 | 28.03 | 27.79 | 27.17 |
| | ActionFormer + VideoMAEv2 [56, 63] | V | 24.80 | 07.25 | 00.77 | 00.09 | 22.26 | 22.23 | 22.16 | 21.70 | 20.71 |
| | ActionFormer + InternVideo [57, 63] | V | 45.57 | 16.07 | 01.75 | 00.23 | 31.78 | 31.77 | 31.73 | 31.56 | 31.14 |
| | BA-TFD [6] | AV | 55.34 | 09.48 | 00.30 | 00.03 | 62.66 | 55.48 | 49.53 | 43.15 | 38.48 |
| | BA-TFD+ [4] | AV | **65.85** | 20.37 | 00.73 | 00.05 | **65.13** | **59.07** | **53.57** | **46.79** | **41.69** |
| | UMMAFormer [65] | AV | 39.07 | **20.77** | **05.62** | **01.16** | 40.39 | 40.19 | 39.51 | 37.53 | 34.93 |
| Subset A | PyAnnote (Zero-Shot) [41] | A | 00.05 | 00.01 | 00.00 | 00.00 | 00.97 | 00.97 | 00.97 | 00.97 | 00.96 |
| | Meso4 [1] | V | 07.13 | 04.17 | 01.45 | 00.39 | 29.34 | 29.34 | 29.27 | 27.58 | 20.54 |
| | MesoInception4 [1] | V | 05.88 | 03.46 | 01.19 | 00.32 | 29.46 | 29.42 | 29.26 | 26.95 | 18.80 |
| | EfficientViT [15] | V | 09.91 | 15.79 | 00.08 | 00.01 | 21.47 | 21.42 | 20.87 | 18.43 | 15.39 |
| | TriDet + VideoMAEv2 [47, 56] | V | 17.45 | 04.01 | 00.24 | 00.02 | 18.47 | 18.43 | 18.28 | 17.53 | 16.02 |
| | TriDet + InternVideo [47, 57] | V | 24.95 | 06.85 | 00.47 | 00.05 | 21.79 | 21.76 | 21.64 | 21.07 | 19.95 |
| | ActionFormer + VideoMAEv2 [56, 63] | V | 16.22 | 03.95 | 00.28 | 00.03 | 18.11 | 18.07 | 17.92 | 17.10 | 15.59 |
| | ActionFormer + InternVideo[57, 63] | V | 30.86 | 09.47 | 00.78 | 00.09 | 24.49 | 24.46 | 24.36 | 23.85 | 22.87 |
| | BA-TFD [6] | AV | 27.79 | 04.31 | 00.12 | 00.01 | 36.71 | 32.50 | 28.82 | 24.02 | 20.58 |
| | BA-TFD+ [4] | AV | 33.23 | 10.07 | 00.36 | 00.03 | 40.54 | 37.07 | 33.63 | 28.50 | 23.82 |
| | UMMAFormer [65] | AV | **68.68** | **40.00** | **11.32** | **02.35** | **51.44** | **51.41** | **51.35** | **51.23** | **50.95** |

Models that have access only to the video-level labels during training and the zero-shot models all perform poorly on this task, except the Face X-Ray and SBI which are designed to be generalizable. Providing the fine-grained segment-level and frame-level labels during training brings an improvement in performance. However, even with the frame-level labels provided during training, the AUC of the best-performing methods is less than 70, due to the multimodal modifications present in AV-Deepfake1M.

The frame- and segment-based deepfake detection methods can only produce frame- and segment-level predictions. Thus, a suitable aggregation strategy is required to generate the video-level predictions. We investigated several popular aggregation strategies, such as *max* (e.g., [6]), *average* (e.g., [15, 22, 55]), and the *average of the highest 5 scores* (e.g., [35]) for video-level predictions. The results of the experiment are presented in Table 9. The results show that *max* is the optimal aggregation strategy on AV-Deepfake1M for the considered deepfake detection methods.

## 5.3 Unimodal Deepfake Detection and Localization

We also evaluated the performance on *subset V* and *subset A*, as described in Section 4.1. As expected, all visual-only methods consistently perform better on *subset V* compared to *fullset* for both temporal localization and detection. The same holds for *subset A* and audio-only methods.

## 5.4 Benchmark Comparison

We conducted additional experiments (Tables 8 and 10) to compare the performance on temporal localization and classification on AV-Deepfake1M and LAV-DF [6].

There is a significant drop in BA-TFD [6] temporal localization performance as compared to LAV-DF (Table 8). A similar pattern is also observed for BA-TFD+ [4] (AP@0.5 96.30 → 44.42) and UMMAFormer [65] (AP@0.5 98.83 → 51.64). For classification (Table 10), the performance of Xception [12], LipForensics [20], Face X-Ray [34], and SBI [48] also drops compared to LAV-DF. These additional results further validate that AV-Deepfake1M is more challenging than LAV-DF.

We conduct the experiments using Xception and BA-TFD pretrained on AV-Deepfake1M then finetune and evaluate on LAV-DF, shown in Table 11. We observe the performance improvements are significant for both temporal localization with BA-TFD and classification with Xception, when compared with models trained on LAV-DF from scratch.

## 6 CONCLUSION

This paper presents AV-Deepfake1M, the largest audio-visual dataset for temporal deepfake localization. The comprehensive benchmark

**Table 7: Deepfake detection benchmark.** Performance comparison of state-of-the-art methods on the proposed AV-Deepfake1M dataset using different evaluation protocols. E5: Ensemble 5.

| Label Access For Training | Methods | Mod. | Fullset | | Subset V | | Subset A | |
|---|---|---|---|---|---|---|---|---|
| | | | AUC | Acc. | AUC | Acc. | AUC | Acc. |
| Zero-Shot | Video-LLaMA (7B) [64] | AV | 50.09 | 25.23 | 50.13 | 33.51 | 50.08 | 33.49 |
| | Video-LLaMA (13B) [64] | AV | 49.50 | 25.02 | 49.53 | 33.35 | 49.30 | 33.36 |
| | Video-LLaMA (7B) E5 [64] | AV | 49.97 | 25.32 | 50.01 | 33.57 | 49.98 | 33.62 |
| | Video-LLaMA (13B) E5 [64] | AV | 50.74 | 25.05 | 50.52 | 33.36 | 50.78 | 33.40 |
| | CLAP [60] | A | 50.83 | 31.99 | 50.91 | 37.83 | 50.67 | 37.54 |
| | M2TR [55] | V | 50.18 | **74.99** | 50.24 | **66.67** | 50.14 | **66.66** |
| | LipForensics [20] | V | 51.57 | 68.84 | 54.37 | 64.13 | 50.65 | 62.19 |
| | Face X-Ray [34] | V | **61.54** | 73.83 | **61.88** | 66.59 | **60.86** | 66.35 |
| | SBI [48] | V | 55.10 | 34.04 | 57.75 | 41.51 | 53.81 | 39.38 |
| Video-level | Meso4 [1] | V | 50.22 | **75.00** | 50.31 | 66.66 | 50.17 | **66.66** |
| | MesoInception4 [1] | V | 50.05 | 75.00 | 50.01 | 66.66 | 50.06 | 66.66 |
| | SBI [48] | V | **65.82** | 69.00 | **67.31** | **67.19** | **65.11** | 65.55 |
| Segment-level | Meso4 [1] | V | 54.53 | 55.83 | 56.81 | **56.78** | 53.34 | 53.89 |
| | MesoInception4 [1] | V | 57.16 | 28.24 | **62.14** | 37.41 | 54.64 | 35.46 |
| | MDS [13] | AV | 56.57 | **59.44** | 54.21 | 53.70 | **56.92** | **58.88** |
| | MARLIN [5] | V | **58.03** | 29.01 | 61.57 | 38.28 | 56.23 | 35.99 |
| Frame-level | Meso4 [1] | V | 63.05 | 49.51 | 76.30 | 64.62 | 56.27 | 47.82 |
| | MesoInception4 [1] | V | 64.04 | 54.13 | 80.67 | 69.88 | 56.28 | 51.73 |
| | Xception [12] | V | **68.68** | 61.33 | **81.97** | **81.39** | **63.19** | 57.45 |
| | EfficientViT [15] | V | 65.51 | **71.80** | 76.74 | 70.89 | 59.75 | **63.51** |

**Table 8: Temporal localization results on the AV-Deepfake1M and LAV-DF.**

| Method | Dataset | AP@0.5 | AP@0.75 | AP@0.95 | AR@50 | AR@20 | AR@10 |
|---|---|---|---|---|---|---|---|
| BA-TFD [6] | LAV-DF [6] | 79.15 | 38.57 | 00.24 | 64.18 | 60.89 | 58.51 |
| | AV-Deepfake1M | 37.37 | 06.34 | 00.02 | 45.55 | 35.95 | 30.66 |
| BA-TFD+ [4] | LAV-DF [6] | 96.30 | 84.96 | 04.44 | 80.48 | 79.40 | 78.75 |
| | AV-Deepfake1M | 44.42 | 13.64 | 00.03 | 48.86 | 40.37 | 34.67 |
| UMMAFormer [65] | LAV-DF [6] | 98.83 | 95.54 | 37.61 | 92.47 | 92.42 | 92.10 |
| | AV-Deepfake1M | 51.64 | 28.09 | 01.57 | 44.07 | 43.45 | 42.09 |

**Table 9: Aggregation strategies.** AUC scores on *fullset* for each method using different aggregation strategies.

| Method → Strategy ↓ | Meso4 [1] | MesoInc4 [1] | Xception [12] | EfficientViT [15] | MARLIN [5] |
|---|---|---|---|---|---|
| max | **63.05** | **64.04** | 68.68 | **65.51** | **58.03** |
| avg | 55.61 | 54.07 | 61.44 | 58.75 | 53.20 |
| avg of top5 | 62.32 | 59.82 | **68.81** | 63.60 | 56.39 |

**Table 10: Performance (AUC ↑) for classification baselines on AV-Deepfake1M and LAV-DF.**

| Label Access | Methods | AV-Deepfake1M | LAV-DF [6] |
|---|---|---|---|
| Zero-shot | LipForensics [20] | 51.57 | 73.34 |
| | Face X-Ray [34] | 61.54 | 69.65 |
| | SBI [48] | 55.10 | 62.84 |
| Video-level | SBI [48] | 65.82 | 67.23 |
| Segment-level | MDS [13] | 56.57 | 82.80 |
| Frame-level | Xception [12] | 68.68 | 83.58 |
| | EfficientViT [15] | 65.51 | 96.50 |

**Table 11: Transfer learning results.** *Dataset for pretraining.*

| Train Data | Methods → Test Data | BA-TFD AP@0.5 ↑ | Xception AUC ↑ |
|---|---|---|---|
| LAV-DF | LAV-DF | 79.15 | 83.58 |
| *AV-Deepfake1M*, LAV-DF | LAV-DF | 83.93 | 90.12 |

of the dataset utilizing state-of-the-art deepfake detection and localization methods indicates a significant drop in performance compared to previous datasets, indicating that the proposed dataset is an important asset for building the next-generation of deepfake localization methods.

**Limitations.** Similarly to other deepfake datasets, AV-Deepfake1M exhibits a misbalance in terms of the number of fake and real videos.

**Broader Impact.** Owing to the diversified and realistic, content-driven fake videos, AV-Deepfake1M will support the development of robust, generalized, audio-visual deepfake detection and localization models.

**Ethics Statement.** We acknowledge that AV-Deepfake1M may raise ethical concerns such as the potential misuse of facial videos of celebrities, and even the data generation pipeline could have a potential negative impact. Misuse could include the creation of deepfake videos or other forms of exploitation. To avoid such issues, we have taken several measures such as distributing the data with a proper end-user license agreement, where we will impose certain restrictions on the usage of the data, such as the data generation technology and resulting content being restricted to research purposes only.

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
