# OpenReview forum: "AV-Deepfake1M: A Large-Scale LLM-Driven Audio-Visual Deepfake Dataset"
_acmmm.org/ACMMM/2024/Conference — MM2024 Oral_

### Official Review · Reviewer_MFmJ · 2024-04-29

**Rating:** 4
**Confidence:** 2

**Summary:**

The researchers present a large-scale temporal deepfake localization task dataset called AV-Deepfake1M.

**Strengths:**

1、The writing in this article is smooth and easy to understand.
2、From the writing of this paper, the work in this paper provides the largest and most challenging dataset to date on the Audio-Visual Deepfake task. It has the potential to contribute to the further development of deepfake localization methods.
3、This paper performs extensive benchmarking on the AV-Deepfake1M dataset.

**Limitations:**

The motivation for this article is unknown. In the introductory section of the article, the researcher claims that most of the existing deepfake benchmark datasets do not include this novel manipulation strategy, so the researcher releases a new large-scale audiovisual dataset, AV-Deepfake1M, to address this gap. And as far as we know, [1] already includes this strategy, has [1] addressed this issue? Admittedly, we might consider this a writing problem.

[1] Z. Cai, K. Stefanov, A. Dhall and M. Hayat, "Do You Really Mean That? Content Driven Audio-Visual Deepfake Dataset and Multimodal Method for Temporal Forgery Localization," 2022 International Conference on Digital Image Computing: Techniques and Applications (DICTA), Sydney, Australia, 2022, pp. 1-10, doi: 10.1109/DICTA56598.2022.10034605.

**Suitability:**

2

---

### Official Review · Reviewer_J9ng · 2024-05-01

**Rating:** 4
**Confidence:** 4

**Summary:**

In this paper, the authors construct a large-scale deepfake dataset based on RE and TTS approaches. In particular, they first modify the transcript of videos with word replacement, and then manipulate the lip movements to align the content. The generated videos are high-quality, with partial manipulation, making detecting these videos a challenging task. The authors conduct extensive benchmark tests on this dataset.

In general, this paper has some merits on deepfake community. Therefore, the reviewer give a positive rating.

**Strengths:**

[+] The proposed dataset is large-scale and high-quality.
[+] The authors conduct extensive experiments on this dataset.

**Limitations:**

[-] Data coverage. Is it appropriate for this dataset to completely disregard FS-related technologies? The reviewer understands that the manipulations discussed in this paper need to ensure the consistency of identity in videos. However, FS, as a potentially dangerous method (especially in video calls), should not be completely ignored. Is it possible to consider applying modified transcripts to some high-quality face-swapped videos to construct other forged subsets?

[-] Assessment of data quality. The videos in the dataset that have only one modality modified, i.e., the audio-modified and video-modified subsets, should exhibit semantic conflicts between the transcripts obtained from speech recognition (e.g., Whisper) and lip reading (e.g., avsr) methods. A higher rate of such conflicts is preferable for these subsets. Conversely, for modifications involving multiple modalities, this rate of conflict should be as low as possible. The authors are advised to include an evaluation metric for this aspect.

**Suitability:**

3

---

### Official Review · Reviewer_4Eyf · 2024-05-08

**Rating:** 4
**Confidence:** 4

**Summary:**

To address  the problem of the localization of small segments of audio-visual manipulations, this paper emulates the process of such content generation and propose the AV-Deepfake1M dataset, including more than 2K subjects, and a total of more than 1M videos.

**Strengths:**

This paper provides a thorough description of the proposed data generation pipeline accompanied by a rigorous analysis of the quality of the generated data. Then, existing deepfake detection and localization methods are evaluated and show that their performances are all a significant drop.

**Limitations:**

(1)How to ensure consistency between audio and video in a synthesized audio-visual dataset? (2) The comprehensive description of multimodal Deepfake forensics methods is incomplete.

**Suitability:**

3

---

### Official Review · Reviewer_jAJa · 2024-05-24

**Rating:** 6
**Confidence:** 4

**Summary:**

This paper proposes the AV-Deepfake1M dataset that contains content-driven (i) video manipulations, (ii) audio manipulations, and iii) audio-visual manipulations for more than 2K subjects resulting in a total of more than 1M videos. The paper provides a thorough description of the proposed data generation pipeline accompanied by a rigorous analysis of the quality of the generated data. The comprehensive benchmark of the proposed dataset utilizing state-of-the-art deepfake detection and localization methods indicates a significant drop in performance compared to previous datasets. The proposed dataset will play a vital role in building the next-generation deepfake localization methods.

**Strengths:**

The proposed AV-Deepfake1M dataset has a significant advantage over the previous dataset LAV-DF in terms of the number of subjects and amount of real and fake videos. Besides, it utilizes LLM to increase content diversity and employs more advanced methods for audio and video generation. Statistical characterization and experimental results demonstrate that this dataset outperforms its predecessor, LAV-DF, in many respects.

**Limitations:**

- The authors can use more TTS models instead of just 2.
- The word "LLM-Driven" only appears once in the title, while in the text "content-driven" is used. It seems better to replace LLM-Driven with content-driven in the title.

**Suitability:**

3

---

### Meta-Review · Area_Chair_9D9b · 2024-07-08

**Recommendation:** Accept (Oral)
**Confidence:** 4

**Metareview:**

All the reviewers are recommending that the paper be accepted. I agree. It does not seem to me to be of sufficient quality to recommend oral presentation, but definitely merits a poster.